# Working memory gates visual input to primate prefrontal neurons

Behrad Noudoost[1]*, Kelsey Lynne Clark[1], Tirin Moore[2]

[1]Department of Ophthalmology and Visual Sciences, University of Utah, Salt Lake City, United States; [2]Department of Neurobiology, and Howard Hughes Medical Institute, Stanford University, Stanford, United States

**Abstract** Visually guided behavior relies on the integration of sensory input and information held in working memory (WM). Yet it remains unclear how this is accomplished at the level of neural circuits. We studied the direct visual cortical inputs to neurons within a visuomotor area of prefrontal cortex in behaving monkeys. We show that the efficacy of visual input to prefrontal cortex is gated by information held in WM. Surprisingly, visual input to prefrontal neurons was found to target those with both visual and motor properties, rather than preferentially targeting other visual neurons. Furthermore, activity evoked from visual cortex was larger in magnitude, more synchronous, and more rapid, when monkeys remembered locations that matched the location of visual input. These results indicate that WM directly influences the circuitry that transforms visual input into visually guided behavior.

## Introduction

Behavior is guided not only by sensory input, but also by information held in working memory (WM). In primates, visually guided eye movements are among the most frequently occurring sensorimotor transformations. Saccadic eye movements occur approximately four to five times each second and require the integration of myriad visual features (e.g., motion and shape) into discrete movements that position visual targets onto the fovea. Furthermore, each movement decision reflects not only visual input, but also information held in WM, such as the behavioral relevance of particular objects and features (*Bichot et al., 2005*; *Hollingworth et al., 2013*; *Hollingworth and Luck, 2009*; *Wolfe and Horowitz, 2004*). How sensory input and WM are integrated in neural circuits to shape behavioral output remains unclear. Studies across multiple species have revealed evidence that WM functions are often associated with networks involved in sensorimotor transformations, including visual-saccadic transformations (*Curtis and D'Esposito, 2006*; *Gnadt and Andersen, 1988*; *Guo et al., 2014*; *Knudsen et al., 1995*; *Knudsen and Knudsen, 1996*; *Kojima et al., 1996*). In these networks, neurons, individually or collectively, often exhibit persistent signaling of information needed to successfully carry out subsequent behaviors. The prevalence of WM-related activity in sensorimotor networks suggests that this may be where WM exerts its influence on sensorimotor transformations. However, the exact mechanism and specific neural circuitry by which WM influences visually guided behaviors are still unknown.

Within primate neocortex, the output of feature-selective neurons in visual cortical areas converges retinotopically onto neurons in the frontal eye field (FEF) (*Schall et al., 1995*), the prefrontal area mostly directly involved in the control of saccades (*Robinson and Fuchs, 1969*; *Schiller et al., 1979*). Neurons within the FEF exhibit functional properties spanning the visual-motor spectrum and also include a substantial portion of neurons with persistent, WM-related activity (*Bruce and Goldberg, 1985*; *Lawrence et al., 2005*). These characteristics make the FEF a likely place to observe an influence of WM on incoming visual signals, particularly given that the FEF transmits a predominantly WM signal to visual cortex (*Merrikhi et al., 2017*). Although much is understood about the role of

*For correspondence:
behrad.noudoost@gmail.com

**Competing interests:** The authors declare that no competing interests exist.

FEF neurons in the control of visually guided saccades (*Schiller et al., 1979*; *Schlag-Rey et al., 1992*; *Tehovnik et al., 2000*), and in the control of visual spatial attention (*Bahmani et al., 2019*; *Gregoriou et al., 2009*; *Moore and Fallah, 2001*), very little is known about how visual, motor, and memory signals are combined within the FEF. Models of FEF microcircuitry generally predict that visual cortical inputs synapse predominantly onto purely visual FEF neurons (e.g., *Heinzle et al., 2007*), yet even this is not known. Furthermore, it is also not known how those visual inputs interact with the current content of WM.

We examined the influence of WM on the efficacy of visual cortical input to the FEF in behaving monkeys. First, we identified FEF neurons with direct input from visual cortex using orthodromic stimulation from extrastriate area V4. Despite the common assumption of visual inputs synapsing onto purely visual FEF neurons, our results revealed that visual cortical input to the FEF instead preferentially targets neurons with both visual and motor properties. Next, we measured the effect of spatial WM on orthodromic activation of FEF neurons and found that the efficacy of visual inputs was enhanced by the memory of spatially corresponding locations. Specifically, the activity evoked in the FEF from visual cortex was larger in magnitude, more synchronous, and more rapid when V4 input matched the location held in WM. These results demonstrate how the content of WM can influence visuomotor transformations in the primate brain.

## Results

We measured the influence of WM on the efficacy of visual cortical inputs to prefrontal cortex in behaving monkeys. Monkeys performed a spatial WM task classically used to characterize FEF

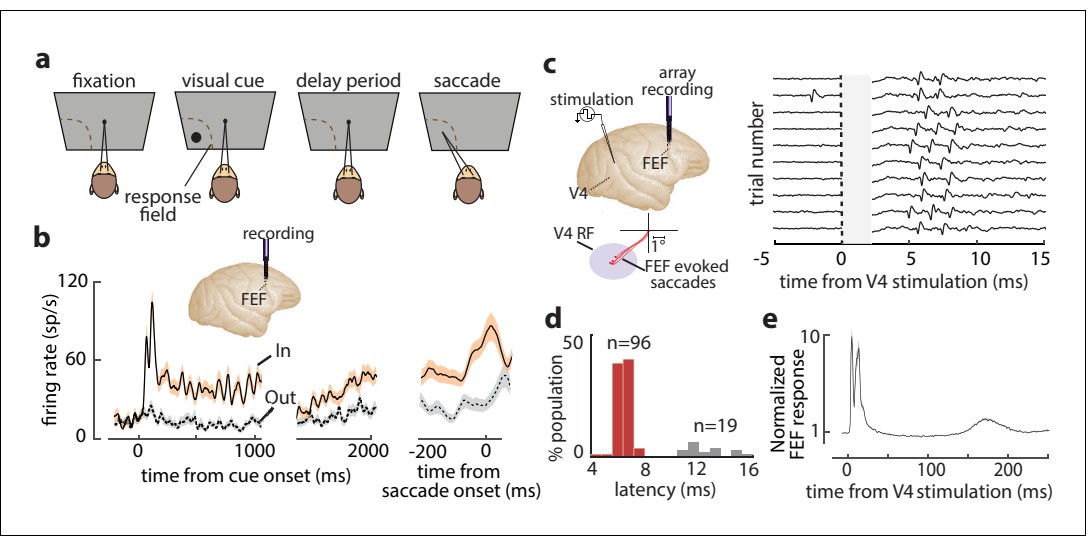

**Figure 1.** FEF responses during the memory-guided saccade (MGS) task and orthodromic activation of FEF neurons from visual cortex. (a) Schematic of the MGS task. Monkey fixates and a visual cue is presented (inside or outside the neuronal response field [RF]). The monkey maintains fixation throughout a delay period, and upon removal of the fixation point, saccades to the remembered location to receive a reward. (b) Response of an example FEF neuron during the MGS task; this neuron shows visual, memory, and motor activity. Responses are shown for cues inside (In, peach) or outside (Out, gray) the RF, aligned to cue onset (left, middle panels) or the saccade (right panel). Traces show mean ± SEM. (c) FEF neurons were orthodromically activated by electrical stimulation of retinotopically corresponding V4 sites (left). Right plot shows evoked spikes from an FEF neuron across 10 trials (stimulation artifact period is shown in gray). (d) Distribution of stimulation-evoked spike latencies for 115 orthodromically activated FEF neurons. (e) Average normalized stimulation-evoked activity of the 96 visual-recipient FEF neurons over time. *Figure 1—figure supplement 1* shows details of stimulation timing and subsequent FEF activity.

The online version of this article includes the following figure supplement(s) for figure 1:

**Figure supplement 1.** Stimulation timing, enhancement, and suppression of visual-recipient FEF neurons' activity following electrical stimulation of V4.

neuronal properties (*Figure 1a*; *Bruce and Goldberg, 1985*; *Lawrence et al., 2005*). *Figure 1b* shows the activity of an example FEF neuron when a monkey remembered a location either inside or outside of the response field (RF; *In* and *Out* conditions, respectively). The neuron responded strongly to a visual cue appearing in the RF and exhibited elevated activity in the delay period when the remembered location coincided with the RF. Prior to saccades to the RF, the neuron also exhibited a burst of motor activity. In primates, direct visual cortical input to prefrontal cortex arrives primarily in the FEF (*Markov et al., 2014a*; *Ungerleider et al., 2008*). We orthodromically activated FEF neurons from retinotopically corresponding sites in extrastriate area V4 (*Figure 1c*) ('Materials and methods'). We targeted overlapping FEF-V4 RFs to maximize the likelihood of finding FEF neurons with input from V4, given the evidence of discrete, retinotopic projections from the latter to the former (*Schall et al., 1995*; *Ungerleider et al., 2008*). Of the 311 single FEF neurons recorded, spikes were reliably elicited by V4 stimulation in 115. Latencies of evoked spikes were bimodally distributed (Hartigan's dip test, $p < 10^{-40}$), with most neurons having latencies <10 ms (n = 96, latency = 6.53 ± 0.67 ms), consistent with monosynaptic transmission (*Figure 1d*; *Gregoriou et al., 2009*; *Nowak and Bullier, 1997*). We focused our analyses on these *visual-recipient* neurons. A smaller population of neurons was activated at longer latencies (n = 19, latency = 12.73 ± 1.40 ms). Visual-recipient neurons exhibited a tri-phasic pattern of evoked activity following orthodromic stimulation (*Figure 1e*, *Figure 1—figure supplement 1*), similar to previous studies employing orthodromic stimulation in primate neocortex (*Matsunami and Hamada, 1984*).

## Characterizing response properties of visual-recipient neurons

We tested whether the properties of visual-recipient neurons differed from those not activated by orthodromic stimulation (*non-activated* neurons) (n = 196). We measured the activity of FEF neurons while monkeys performed a spatial WM task that temporally dissociates visual, memory, and motor components of neuronal responses ('Materials and methods'). *Figure 2a* shows the activity of an example visual-recipient FEF neuron when the monkey remembered a location either inside or outside of the RF. This neuron responded strongly to a visual cue appearing in the RF but exhibited minimal activity in the delay period, and it was not selective for the remembered location. Prior to saccades to the RF, the neuron exhibited a burst of motor activity. Thus, this neuron exhibited visual and motor, not memory-related, activity. We compared the proportions of neurons with significant visual, memory, and motor activity between the visual-recipient and non-activated neuronal populations. Each component of activity was measured as the significant response enhancement in the corresponding behavioral epoch during the In condition (see 'Materials and methods'). We found that visual-recipient FEF neurons exhibited greater proportions of visual ($\chi^2 = 9.42$, p = 0.002) and motor activity ($\chi^2 = 10.71$, p = 0.001) than non-activated neurons. However, the proportion of neurons with memory activity did not differ between the two populations ($\chi^2 = 0.99$, p = 0.318) (*Figure 2b*, left). Next, we asked whether the greater proportion of neurons with visual and motor activity among visual-recipient neurons corresponded to a larger proportion of visuomotor neurons. To test that, we compared the proportion of visuomotor neurons between the visual-recipient and non-activated neurons with significant selectivity between the In and Out conditions during the visual and/or motor epochs (see 'Materials and methods') (n = 74, non-activated; n = 59, visual-recipient). Overall, the relative proportions of visual, visuomotor, and motor neurons differed between the two populations ($\chi^2 = 6.89$, p = 0.0319), with a higher prevalence of visuomotor neurons among the visual-recipient population (66 vs 44%, $\chi^2 = 11.34$, $p < 10^{-3}$), and a lower proportion of purely visual (19 vs 31%, $\chi^2 = 4.39$, p = 0.036) and purely motor neurons (15 vs 26%, $\chi^2 = 4.23$, p = 0.039) (*Figure 2b*, right). Thus, the increased prevalence of visual and motor selectivity among the visual-recipient neurons reflected a larger proportion of visuomotor neurons.

We considered that the larger motor signals among visual-recipient neurons could have resulted from differences in the alignment of the cue stimulus with the centers of FEF visual and movement fields, as they can be significantly misaligned (*Bruce and Goldberg, 1985*; *Schafer and Moore, 2011*). Thus, we measured the magnitude of motor activity (selectivity to In vs Out) across varying amounts of visual activity in the two populations of neurons ('Materials and methods') (*Figure 2c*, left). This comparison revealed that for a given level of visual activity, visual-recipient neurons exhibited a larger component of motor activity than non-activated neurons (ANCOVA, F = 10.15, p = 0.002). In contrast, a corresponding comparison of memory and visual activity in the two populations revealed no differences (ANCOVA, F = 0.23, p = 0.631) (*Figure 2c*, right). Thus,

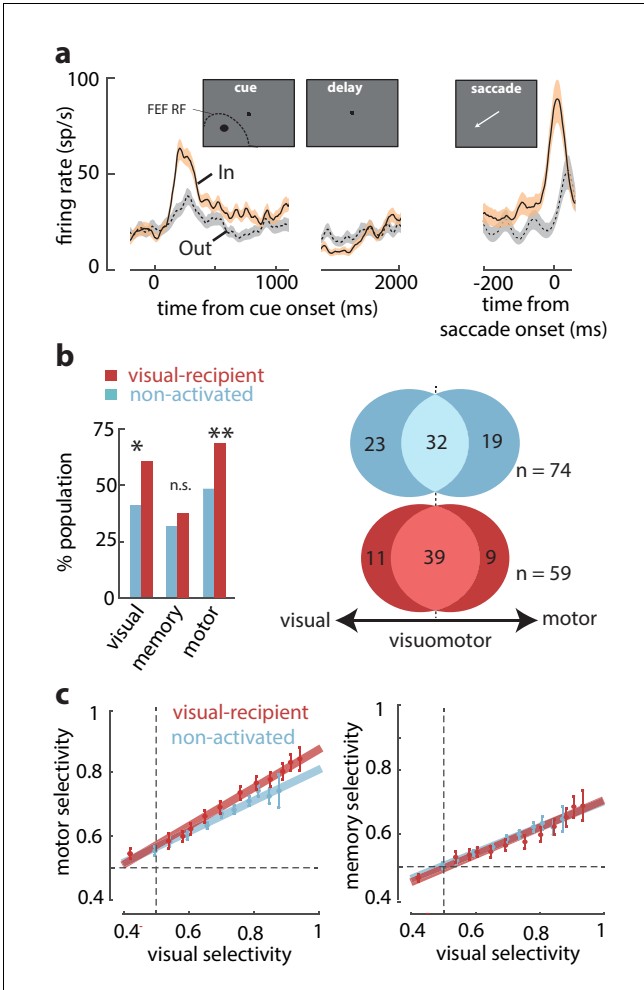

**Figure 2.** Overrepresentation of visuomotor activity in visual-recipient FEF neurons. (a) Activity of an example visual-recipient FEF neuron when the cue appeared In (peach) or Out (gray) of the RF. Plots show firing rates aligned to the onset of the visual cue (left), offset of the visual cue (middle), and to saccade onset (right). (b) Left: percent of the population exhibiting visual, memory, and motor activity for visual-recipient (cardinal) and non-activated (turquoise) FEF neurons. *p<0.05, **p<0.001; ns denotes p>0.05. Right: Venn diagrams showing the numbers of neurons with significant selectivity in the visual, motor, or both epochs for non-activated (n = 74) and visual-recipient (n = 59) neurons. *Figure 2—figure supplement 1* compares the distributions of visual, memory, and motor selectivity between visual-recipient, non-activated, and slow-input FEF populations; *Figure 2—figure supplement 2* provides statistical comparisons of selectivity in each period between these populations. (c) Motor selectivity (left) and memory selectivity (right) as a function of visual selectivity for visual-recipient and non-activated FEF neurons.

The online version of this article includes the following figure supplement(s) for figure 2:

**Figure supplement 1.** Distribution of visual, memory, and motor selectivity for random subsamples of non-input (blue) and visual-recipient (red) FEF neurons.

**Figure supplement 2.** Statistical comparisons of the visual, memory, and motor selectivity for non-activated, slow-input, and visual-recipient FEF neurons.

whereas memory signals among visual-recipient FEF neurons were equal to those of non-activated neurons, motor signals were significantly overrepresented (see *Figure 2—figure supplements 1* and *2*). It should be noted, however, that in spite of the anatomical evidence of retinotopic projections from V4 to FEF (*Schall et al., 1995*; *Ungerleider et al., 2008*), one cannot rule out the possibility that non-activated FEF neurons receive inputs from V4 neurons with non-overlapping RFs. Those inputs could be distributed onto FEF neurons differently from those arriving retinotopically.

Nonetheless, our results show that FEF neurons receiving retinotopic inputs from V4 exhibit stronger motor activity than those without such inputs.

## WM alters efficacy of V4 input to FEF

In contrast to the disproportionate prevalence of motor signals among neurons receiving input from visual cortex, reciprocal projections of the FEF to visual cortex originate disproportionately from neurons with memory-related activity (*Merrikhi et al., 2017*). This implicates the FEF as a possible source of the observed memory-dependent modulation in visual cortex (*Bahmani et al., 2018*; *Supèr et al., 2001*; *van Kerkoerle et al., 2017*). It also suggests that memory-related signals may interact with the efficacy of visual inputs arriving in prefrontal cortex. To test this possibility, we compared the efficacy of orthodromic activation of FEF neurons from area V4 between WM for different spatial locations. Previous studies have shown that the efficacy of orthodromic activation of primary visual cortex neurons from the thalamus (*Briggs et al., 2013*), and of extrastriate cortex from primary visual cortex (*Ruff and Cohen, 2016*), are both altered during spatial attention. Using a similar approach, we compared the activity evoked from visual cortex by orthodromic stimulation during the delay period of the MGS task when monkeys remembered different cue locations.

We examined the influence of WM across the full population of 96 visual-recipient neurons. On average, the proportion of stimulation-evoked spikes increased by 19% when monkeys remembered locations inside the RF, compared to outside (Spike count$_{In}$ = 0.22 ± 0.005, Spike count$_{Out}$ = 0.18 ± 0.004; $p<10^{-6}$) (*Figure 3—figure supplement 1*). For each neuron, we measured the evoked response magnitude to quantify the efficacy of stimulation (see 'Materials and methods') and compared the magnitude between trials with different memory locations. The magnitude of evoked responses was significantly greater when monkeys remembered cue locations inside the RF, compared to outside (Response magnitude$_{In}$ = 0.77 ± 0.04, Response magnitude$_{Out}$ = 0.68 ± 0.03; $p<10^{-10}$) (*Figure 3a*). The increase in efficacy was independent of the delay period selectivity of FEF neurons (*Figure 3—figure supplement 2*; r = 0.12, p = 0.235, Pearson correlation). Thus, the efficacy of visual input to the FEF depended on the content of WM. In addition, we found that the latency of evoked spikes was slightly reduced when monkeys remembered locations inside, compared to outside, the RF (Latency$_{In}$ = 7.88, Latency$_{Out}$ = 8.04; $p<10^{-10}$) (*Figure 3b*). *Figure 3c* shows the spikes evoked from two example FEF neurons in response to V4 stimulation during the memory period. Evoked spikes from one neuron increased by ~30% during memory of locations inside, compared to outside the RF. The number of evoked spikes from a second, simultaneously recorded, neuron was similar in the two RF conditions, but spike onset appeared more rapid during the In condition, consistent with the latency effects. As a consequence, when combined, the evoked spikes of the two neurons were more synchronous during memory of locations inside of the RF (*Figure 3c*) ('Materials and methods'). We compared the proportion of joint spikes in all simultaneously recorded pairs of visual-recipient neurons (n = 509) across the different memory locations. Overall, we found that the proportion of joint spiking, when controlled for firing rate (see 'Materials and methods'), increased by nearly 60% during memory of locations inside, compared to outside the RF (Prob$_{In}$ = 0.103 ± 0.002, Prob$_{Out}$ = 0.065 ± 0.001;, $p<10^{-63}$) (*Figure 3d*). Thus, activity evoked in the FEF from visual cortex was larger, more synchronous, and more rapid when monkeys engaged WM at RF locations.

## Discussion

We found that visual inputs to the FEF disproportionately drove neurons with both visual and motor properties. Rather than exhibiting purely visual properties, as might be expected (*Sato and Schall, 2003*), visual-recipient FEF neurons also signaled the direction of impending eye movements. Although surprising, this result seems consistent with the pattern of visual cortical connections with the FEF (*Barone et al., 2000*; *Markov et al., 2014b*). V4 inputs terminate in all layers of the FEF (*Ungerleider et al., 2008*), thus potentially distributing inputs across different functional classes of neurons. The bias in those inputs toward visuomotor neurons indicates that rather than being integrated at a subsequent processing stage within the FEF, as proposed by models of FEF microcircuitry (*Brown et al., 2004*; *Heinzle et al., 2007*), sensory and motor signals are combined at the interface between visual and prefrontal cortex. Moreover, this result is consistent with the observation of equal visual latencies between visuomotor and visual FEF neurons (*Schall, 1991*), and the

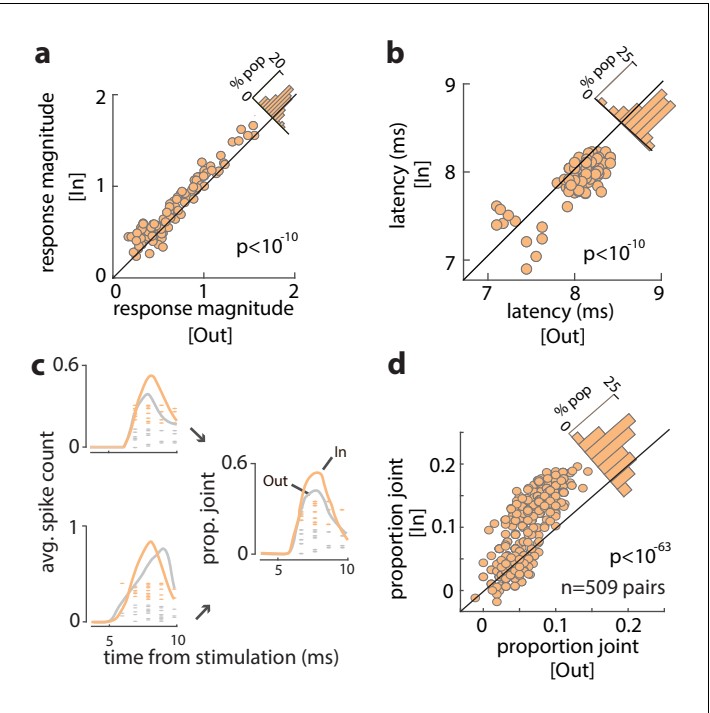

**Figure 3.** Increased efficacy of visual input to the FEF during WM. (a) Magnitude of orthodromically evoked FEF responses and (b) latency of orthodromically evoked spikes during the delay period of the WM task, for memory of locations inside vs outside of the RF, for all visual-recipient FEF neurons. *Figure 3—figure supplement 1* shows the average spike counts for In vs Out, and *Figure 3—figure supplement 2* shows the relationship between change in efficacy and delay selectivity. (c) Left: mean spike counts and raster plots following V4 stimulation for two example neurons during memory of locations inside (peach) and outside (gray) of the RF. Right: proportion of joint spikes in the two example neurons during the two memory conditions. (d) Mean proportion of joint spikes for all pairs of visual-recipient FEF neurons during the two memory conditions, adjusted for firing rate. All data are from stimulation during the delay period of the MGS task.

The online version of this article includes the following figure supplement(s) for figure 3:

**Figure supplement 1.** Average spike count as a measure of stimulation efficacy across the population of visual-recipient FEF neurons, for memory Out (x-axis) vs In (y-axis); diagonal histogram shows differences.
**Figure supplement 2.** Working memory-induced change in stimulation efficacy as a function of the FEF neuron's delay period selectivity.

finding that visuomotor neurons exhibit greater visual discrimination than visual neurons during certain saccade tasks (*Costello et al., 2013*).

More importantly, we found that the engagement of WM at RF locations increased the efficacy of visual input to FEF neurons. When monkeys remembered RF locations, activity evoked in the FEF from visual cortex was larger in magnitude, more synchronous, and more rapid. The increased efficacy of visual input to the FEF was independent of delay period activity, and it was observed whether or not neurons exhibited WM-related activity. Given that visual cortical inputs appear to drive neurons across functional classes, this result suggests that those inputs interact constructively with WM-related signals within the FEF. Similarly, previous work has shown that the deployment of attention increases the input efficacy of thalamocortical (*Briggs et al., 2013*) and corticocortical (*Ruff and Cohen, 2016*) connections, and, thus, our results indicate that attention and WM exert analogous influences on visual input across multiple levels of the visual hierarchy. In both cases, the cellular-level mechanisms of the observed enhancement remain to be determined, as they could result either from a pre-synaptic or from a post-synaptic facilitative mechanism, or both. Moreover, those mechanisms may differ in important ways between attention and WM.

Evidence from multiple species suggests that top-down attention arises from biases in the selection of sensory input based on the content of WM (*Desimone and Duncan, 1995*; *Knudsen, 2007*; *Miller and Cohen, 2001*; *Soto et al., 2010*). A recent study indicates that FEF neurons with WM-

related activity disproportionately provide input to neurons in area V4 (*Merrikhi et al., 2017*) and thus underlie the FEF's contribution to the modulation of visual cortical activity classically observed during spatial attention (*Ekstrom et al., 2009*; *Gregoriou et al., 2014*; *Moore and Armstrong, 2003*). In contrast to visual cortical projections from the FEF, inputs to the FEF did not preferentially target memory-related neurons, but instead neurons with visuomotor properties. This suggests that interactions between the FEF and visual cortex are not strictly recurrent (*Knudsen, 2007*; *Noudoost et al., 2014*) and that memory activity within the FEF, rather than reinforcing its own content, may instead facilitate the transformation of visual inputs into motor commands. Combined, our results suggest a basis for the well-documented interdependence of attention, WM, and gaze control (*Ikkai and Curtis, 2011*; *Jonikaitis and Moore, 2019*), which at the circuit level remains an important puzzle to solve.

## Materials and methods

Two adult male rhesus monkeys (*Macaca mulatta*) were used in this study. All experimental procedures were in accordance with National Institutes of Health *Guide for the Care and Use of Laboratory Animals*, the Society for Neuroscience Guidelines and Policies, and Stanford University Animal Care and Use Committee.

### General and surgical procedures

Each animal was surgically implanted with a head post, a scleral eye coil, and two recording chambers. Two craniotomies were performed on each animal, allowing access to dorsal V4, on the prelunate gyrus, and FEF, on the anterior bank of the arcuate sulcus. Eye position monitoring was performed via the scleral search coil and was digitized at 500 Hz (CNC Engineering). Eye monitoring, stimulus presentation, data acquisition, and behavioral monitoring were controlled by the CORTEX system. Visual stimuli presented to estimate V4 RFs were 1.2–1.9°×0.2–0.4° bar stimuli appearing at four possible orientations (0, 45, 90, and 135°). All stimuli were presented on a 29°×39° (22″) colorimetrically calibrated CRT monitor (Mitsubishi Diamond Pro 2070SB-BK) with medium short-persistence phosphors (refresh rate 77 Hz).

### Neurophysiological recording and data acquisition

Neurophysiological recordings of single neurons in awake monkeys were made through two surgically implanted cylindrical titanium chambers (20 mm diameter) overlaying the prelunate gyrus (V4) and the pre-arcuate gyrus (FEF). Electrodes were lowered into the cortex using a hydraulic microdrive (Narishige). Neural activity was recorded extracellularly with varnish-coated tungsten microelectrodes (FHC) of 0.2–1.0 MΩ impedance (measured at 1 kHz) in V4, and via linear electrode array (v-probe, Plexon) in FEF. Extracellular waveforms were digitized and classified as single neurons using both template-matching and window discrimination techniques (FHC, Plexon). Area V4 was identified based on stereotaxic location, position relative to nearby sulci, patterns of gray and white matter, and response properties of units encountered; the FEF was identified based on these factors and the ability to evoke fixed-vector eye movements with low-current electrical stimulation. Prior to beginning data collection, the location of FEF and V4 within the recording chambers was established via single-electrode exploration.

#### Eye calibration

Each day began by calibrating the eye position; once the electrode was positioned in the FEF, the same task was used with stimulation to verify that the electrode was in FEF and estimate the RF center. The fixation point, an ~1 degree of visual angle (d.v.a.) white circle, appeared in the center of the screen, and the monkey maintained fixation within a ±1.5 d.v.a. window for 1.5 s. For eye calibration, no stimulation was delivered, and the fixation point could appear either centrally or offset by 10 d.v.a. in the vertical or horizontal axis.

#### Achieving FEF-V4 RF overlap

In each recording session, we first localized sites within the FEF and V4 where neurons exhibited retinotopically corresponding representations, meaning that V4 RFs overlapped with the end point of

saccade vectors evoked by FEF microstimulation (*Merrikhi et al., 2017*; *Moore and Armstrong, 2003*). Preliminary RF mapping in V4 was conducted while the monkey fixated within a ± 1.5 d.v.a. window around the central fixation point, while ~2.5 × 4 d.v.a. white bars swept in eight directions (four orientations) across the approximate location of the neuron's RF. Responses from the recording site were monitored audibly and visually by the experimenter, and the approximate boundaries of the RF were noted for the positioning of stimuli in subsequent behavioral tasks. To establish that the electrode was positioned within the FEF and to estimate the FEF RF location, microstimulation was delivered randomly on 50% of trials while the animal performed a passive fixation task. Microstimulation consisted of trains (50–100 ms) of biphasic current pulses (≤50 µA; 250 Hz; 0.25 ms duration). On no-stimulation trials, the monkey was rewarded for maintaining fixation; on stimulation trials, the monkey was rewarded whether fixation was maintained or not following microstimulation. The ability to evoke saccades with low stimulation currents (≤50 µA) confirmed that the electrode was in the FEF; the end point of the stimulation-evoked saccades provided an estimate of the RF center for the FEF site.

### MGS task

The FEF visual, motor, and delay activity were characterized in an MGS task. Monkeys fixated within a ±1.5 d.v.a. window around the central fixation point. After 1 s of fixation, a 1.35 d.v.a. square cue was presented and remained onscreen for 1 s and was then extinguished. The animal then remembered the cue location while maintaining fixation for an additional 1 s (delay period) before the central fixation point was removed. The animal then had 500 ms to shift its gaze to a ±4 d.v.a. window around the previous cue location and remain fixating there for 200 ms to receive a reward. This task was performed with two potential cue locations, located at 0° and 180° relative to the estimated RF center.

### Electrical stimulation

During the MGS task described above, electrical stimulation was delivered to V4 during the fixation, visual, delay, or saccade period on 50% of trials (on the other 50% of trials, there was no stimulation). For identifying antidromically and orthodromically activated FEF neurons (see below), and evaluating stimulation efficacy, electrical stimulation used consisted of a single biphasic current pulse (600–1000 µA; 0.25 ms duration, positive phase first). Stimulation times were 500 ms after initiating fixation (fixation), 500 ms after visual cue onset (visual), 500 ms after cue offset (delay), or 150 ms after the go cue (saccade). Data from the visual, delay, and saccade stimulation periods were used to identify orthodromically activated FEF neurons; data from stimulation during the delay period were used to measure the impact of WM on visual input efficacy.

## Statistical analysis

### Latency of stimulation-evoked spikes

The probability of firing in each 1 ms bin following V4 stimulation was measured for stimulation trials and compared to the probability of firing in a time-matched window from non-stimulation trials. The first bin in which the probability of firing was significantly greater for stimulation trials was designated the latency of stimulation-evoked spikes. Hartigan's dip test was used to test the bimodality of the latency distribution (*Figure 1d*).

### Identifying orthodromically activated neurons

Electrical stimulation of V4 evoked spikes in FEF via both orthodromic and antidromic stimulation. Antidromically evoked spikes (in V4-projecting FEF neurons) were of short latency and were confirmed via the collision test (using stimulation data collected during the MGS task described above). This test identifies antidromically activated neurons: when V4 stimulation was delivered within a few milliseconds of a spontaneously generated spike from a recorded FEF neuron, spikes artificially evoked from that neuron by V4 microstimulation were eliminated. Orthodromically activated neurons will still have an evoked spike in this period following a spontaneously generated spike. The characteristics of FEF neurons antidromically activated by V4 stimulation have been reported elsewhere (*Merrikhi et al., 2017*).

## Assessment of stimulation-evoked activity

The efficacy of stimulation was compared during the delay period of the WM task, while monkeys remembered a cue location inside or outside of a neuron's RF. All responses were measured within the 5–9 ms post-stimulation period, to stay consistent with the latency of visual-recipient neurons as shown in *Figure 1d*. To focus on stimulation-evoked spikes, rather than spontaneous spiking activity, all measures were adjusted by subtracting the same measure (firing rate, spike count, or probability) observed during the same time period on non-stimulated trials. *Figure 3a and d* shows adjusted values, following subtraction of the same measure during the non-stimulated trials. The data shown in *Figure 3b* (neuronal latency) and *Figure 3c* (spiking activity of an example neuronal pair) are raw, non-adjusted values. The evoked response magnitude (*Figure 3a*) was calculated based on the log ratio of spike counts before vs after stimulation, and subtracting the same measure during non-stimulation trials: $\log_{10}(\text{resp after/resp before})_{STIM} - \log_{10}(\text{resp after/resp before})_{NONSTIM}$. The proportion of joint spiking (*Figure 3d*) was measured as the proportion of trials in which spikes from both neurons in a pair occurred within 1 ms and was averaged between 5 and 9 ms post-stimulation. Higher firing rates will increase the probability of joint spiking. To control for firing rate, we subtracted off the proportion of joint spikes occurring in trials shuffled within each condition from that measured during non-shuffled trials. Similar to other measures, this value in non-stimulated trails was also subtracted from that in the stimulated trials. Thus, the reported joint proportions are controlled for changes in firing rate due to both stimulation and WM, and measures only the change in synchronous firing.

## Characterizing FEF response properties

The visual, motor, and delay period activity of FEF neurons was measured using the MGS task described above. The visual period included activity 100–1000 ms after stimulus onset. Delay period activity was measured from 300 to 1000 ms after stimulus offset. Motor activity was quantified in the presaccadic window 125 ms before the saccade onset. These time windows were also used to quantify the different types of activity using a receiver-operating characteristic (ROC) analysis to measure selectivity, as described below. When determining whether a neuron exhibited significant visual or delay activity, activity in the visual and delay periods of the In condition was compared to the activity of the same neuron during fixation (300 ms before stimulus onset), using the Wilcoxon sign-rank test ($p < 0.05$). When determining whether a neuron had significant motor activity, saccade-aligned activity in the In condition was compared to saccade-aligned activity earlier in the trial (450–250 ms before saccade onset), using the sign-rank test ($p < 0.05$).

The strength of visual, memory, and motor activity was measured as the selectivity of neurons to the In and Out conditions during the visual, delay, and motor epochs, respectively, and was quantified using an ROC analysis. This method compared the distributions of firing rates for trials in which the memory cue appeared inside versus outside the neuron's RF (*Green and Swets, 1966*). The areas under ROC curves were used as a measure of selectivity for cue location and were calculated as in previous studies (*Armstrong and Moore, 2007*; *Britten et al., 1992*). Specifically, we computed the average firing rate in the visual, delay, and saccade windows defined above, for In and Out trials. We then computed the probability that the firing rate in each stimulus condition exceeded a criterion. The criterion was incremented from 0 to the maximum firing rate, and the probability of exceeding each criterion was computed. Thus, a single point on the ROC curve is produced for each increment in the criterion, and the entire ROC curve is generated from all of the criteria. The area under the ROC curve is a normalized measure of the separation between the two firing-rate distributions obtained when the WM cue appeared inside versus outside the neuronal RF and provides a measure of how well the neuronal response discriminates between the two conditions.

## Acknowledgements

This work was supported by National Institutes of Health grants R01EY02694, R01NS113073, and R01MH121435 to BN, an unrestricted grant from Research to Prevent Blindness, Inc to Moran Eye Center, University of Utah, and R01EY014924 and Howard Hughes Medical Institute grants to TM.

## Additional information

### Funding

| Funder | Grant reference number | Author |
|---|---|---|
| National Eye Institute | R01EY014924 | Behrad Noudoost Tirin Moore |
| National Eye Institute | R01EY02694 | Behrad Noudoost |
| National Institute of Neurological Disorders and Stroke | R01NS113073 | Behrad Noudoost |
| National Institute of Mental Health | R01MH121435 | Behrad Noudoost |

The funders had no role in study design, data collection and interpretation, or the decision to submit the work for publication.

### Author contributions

Behrad Noudoost, Conceptualization, Data curation, Formal analysis, Validation, Visualization, Methodology, Writing - original draft, Writing - review and editing; Kelsey Lynne Clark, Data curation, Formal analysis, Validation, Writing - original draft, Writing - review and editing; Tirin Moore, Conceptualization, Supervision, Funding acquisition, Validation, Visualization, Writing - original draft, Project administration, Writing - review and editing

### Author ORCIDs

Behrad Noudoost (iD) https://orcid.org/0000-0002-1588-027X
Kelsey Lynne Clark (iD) https://orcid.org/0000-0002-1791-3960
Tirin Moore (iD) https://orcid.org/0000-0002-3345-2930

### Ethics

Animal experimentation: All experimental procedures were in accordance with National Institutes of Health Guide for the Care and Use of Laboratory Animals, the Society for Neuroscience Guidelines and Policies, and Stanford University Animal Care and Use Committee (protocol #9900).

### Decision letter and Author response

Decision letter https://doi.org/10.7554/eLife.64814.sa1
Author response https://doi.org/10.7554/eLife.64814.sa2

## Additional files

### Supplementary files

• Transparent reporting form

### Data availability

The source Plexon datafile and the readme file on how to analyze the data are now publicly available via the repository.

The following dataset was generated:

| Author(s) | Year | Dataset title | Dataset URL | Database and Identifier |
|---|---|---|---|---|
| Noudoost B, Moore T | 2021 | FEF MGS with V4 stim | https://gin.g-node.org/KClark/collision | GIN, 10.12751/g-node.b0lth8 |

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
