## [Decision Letter]

**Acceptance summary:**

This study documented the properties of neurons in the frontal eye field (FEF), a cortical brain area traditionally thought to receive visual input and transform that input into motor commands. The authors used extracellular recordings and electrical microstimulation in behaving non-human primates (NHPs) to add to this view by showing that visual input to FEF from visual area V4 appears to be gated by working -memory activity in FEF. This kind of circuit-breaking is rare and valuable in NHPs and provides new insights into mechanisms that subserve the flexible, context-dependent flow of information in the primate brain.

**Decision letter after peer review:**

Thank you for submitting your article "Working Memory Gates Visual Input to Primate Prefrontal Neurons" for consideration by *eLife*. Your article has been reviewed by 2 peer reviewers, and the evaluation has been overseen by Joshua Gold as the Reviewing and Senior Editor. The following individual involved in review of your submission has agreed to reveal their identity: Shengtao Yang (Reviewer #1).

Essential revisions:

1. Orthodromic activation of FEF neurons via V4 stimulation increases the percentage of FEF events that lead to spikes and decreases their latency during working memory. Such an effect appears expectable if FEF neurons are at a higher level of when a stimulus in their receptive field is held in memory compared to a stimulus out of their receptive field. Are the authors suggesting something special about working memory? Would the same outcome not be expected during fixation or smooth pursuit for FEF neurons that are activated by these states? It was not clear that efficacy of transmission itself improves by working memory, just the likelihood that the spiking threshold would be reached. This was considered a major concern.

2. It would strengthen the authors thesis to discuss the existing functional evidence (in addition to anatomical evidence) that motor FEF neurons receive visual input and can plan movements accordingly. See for example Costello et al. J. Neurosci 2013, 33(41):16394-408.

3. The authors match the receptive location of FEF and V4 neurons to maximize the chances of identifying monosynaptically connected neurons between the two areas. However, a negative finding of no effects of orthodromic activation does not entirely rule out that the FEF neuron under study receive V4 input, from another site. Some discussion is warranted on this point.

4. In classical working memory tasks, the task periods usually consist of fixation, cue, delay and then response period. The neural activity during delay period is considered as working memory related signal. However, in current study, the authors didn't point out whether only delay period activity was included in analysis when they compared synaptic efficacy between stimulation and non-stimulation trials, in Figure 4a. Because the differences of neuronal response during fixation period cannot be viewed as relevant to information held in working memory, it may be better if only neuronal activity in delay period was included in their analysis.

Relatedly, did all of the 96 visual-recipient FEF neurons show working memory-related activity in the memory-guided saccade task? The example neuron in Figure 3a didn't show a significant difference between In and Out trials during delay period. If the visual-recipient neuron didn't show working memory-related activity, what is the basis for the claim that enhanced synaptic efficacy from V4 to FEF was caused by working memory?

5. Did the two example neurons in Figure 4c show adjusted values (subtracting the same measure during non-stimulated trials)? The authors mentioned in Method that Figure 4 showed adjusted values, but it may not be applicable for raster plot in Figure 4c. It may be helpful that using adjusted values show stimulation effects on evoked spike counts during memory In and Out trials.

6. Did the authors find some FEF cells showing elevated firing during delay period in outside-RF trials compared with baseline firing? These elevated firings would not be caused by the RF cue and thus could more readily be interpreted as a working memory signal.

7. The sample size should be indicated in the Figure 3b Venn diagram.

8. It would be useful to indicate the electrical stimulus protocol in Figure 1.

*Reviewer #1:*

The authors of Working Memory Gates Visual Input to Primate Prefrontal Neurons studied how working memory influence information transmitting from V4 to frontal eye field via extracellular recording and electrical stimulation on behaving primate. They found that V4 neurons target FEF neurons with both visual and motor property, and its synaptic efficacy of V4 to FEF was enhanced by working memory. These findings are interesting and important to our understanding about how our brain acts during daily WM-related activity.

*Reviewer #2:*

This is a very interesting study, examining the properties of different types of neurons in the primate Frontal Eye Fields. It is commonly assumed that a serial processing of information takes place in the frontal lobe, from visual representation, to working memory maintenance, to motor output. However, some evidence to the contrary has also been reported, creating a debate in the field. The authors have characterized meticulously FEF neurons receiving V4 projections, by means of orthodromic stimulation. They report two main findings: that visual-input recipient neurons in FEF exhibit substantial motor activity and that working memory alters the efficacy of V4 input to FEF. The paper provides an important addition to our understanding of FEF processing. Although the first result is unambiguous, and goes against the traditional view of the FEF, the interpretation of the second is less straightforward and would need to be qualified further.

1. Orthodromic activation of FEF neurons via V4 stimulation increases the percentage of FEF events that lead to spikes and decreases their latency during working memory. Such an effect appears expectable if FEF neurons are at a higher level of when a stimulus in their receptive field is held in memory compared to a stimulus out of their receptive field. Are the authors suggesting something special about working memory? Would the same outcome not be expected during fixation or smooth pursuit for FEF neurons that are activated by these states? It was not clear that efficacy of transmission itself improves by working memory, just the likelihood that the spiking threshold would be reached.

2. It would strengthen the authors thesis to discuss the existing functional evidence (in addition to anatomical evidence) that motor FEF neurons receive visual input and can plan movements accordingly. See for example Costello et al. J. Neurosci 2013, 33(41):16394-408.

3. The authors match the receptive location of FEF and V4 neurons to maximize the chances of identifying monosynaptically connected neurons between the two areas. However, a negative finding of no effects of orthodromic activation does not entirely rule out that the FEF neuron under study receive V4 input, from another site. Some discussion is warranted on this point.

---

## [Author Response]

Essential revisions:1. Orthodromic activation of FEF neurons via V4 stimulation increases the percentage of FEF events that lead to spikes and decreases their latency during working memory. Such an effect appears expectable if FEF neurons are at a higher level of when a stimulus in their receptive field is held in memory compared to a stimulus out of their receptive field. Are the authors suggesting something special about working memory? Would the same outcome not be expected during fixation or smooth pursuit for FEF neurons that are activated by these states? It was not clear that efficacy of transmission itself improves by working memory, just the likelihood that the spiking threshold would be reached. This was considered a major concern.

We agree that this is an important point worth addressing more clearly. Thus, we clarify in the revised paper in several ways that the observation of enhanced efficacy of orthodromic activation of FEF neurons is not necessarily expected. First, as described in the methods, and clarified further in the revision, evoked spiking responses are always measured as the change from the non-stimulation condition during the same time period. So response magnitude is normalized to this non-stimulation baseline, and thus one could have reasonably expected there to be no change in the evoked activity between the IN and OUT conditions if the combined (stimulation and endogenous) inputs are added linearly. Second, as we clarify in the revised paper, the enhancement effect is observed in neurons whether or not they exhibit delay activity. Indeed, neurons with equal spike rates in the IN and OUT conditions exhibit equally large enhancement effects. To make this point clearer, we have added a supplemental analysis and figure (Figure 3, Supplement 2) that is similar to that used previously in studies demonstrating similar increased efficacy with attention (see Supplementary Figure 1d from Briggs et al., 2013). This analysis shows no relationship between the change in efficacy and delay selectivity (i.e. difference in firing rate between IN and OUT conditions). So, whether or not changes in efficacy are also present during changes in ‘states’ other than WM (it appears so for attention, and could also be so during smooth pursuit), it is clear that the observed change is not merely due to ‘a higher level’ of activity in the FEF neurons. Note also that, on the contrary, we do not argue that WM is special (in the discussion). Instead, we draw parallels with the effects seen during attention and note other evidence of similar underlying mechanisms.

2. It would strengthen the authors thesis to discuss the existing functional evidence (in addition to anatomical evidence) that motor FEF neurons receive visual input and can plan movements accordingly. See for example Costello et al. J. Neurosci 2013, 33(41):16394-408.

We thank the reviewers for the suggestion, and have added the following text to the Discussion section:

“Moreover, this result is consistent with the observation of equal visual latencies between visuomotor and visual FEF neurons (Schall, 1991), and the finding that visuomotor neurons exhibit greater visual discrimination than visual neurons during certain saccade tasks (Costello et al., 2013).”

3. The authors match the receptive location of FEF and V4 neurons to maximize the chances of identifying monosynaptically connected neurons between the two areas. However, a negative finding of no effects of orthodromic activation does not entirely rule out that the FEF neuron under study receive V4 input, from another site. Some discussion is warranted on this point.

We have revised the wording in the results (pages 5 and 8) to clarify that we targeted overlapping FEF-V4 receptive fields based on the fact that projections from V4 to FEF appear to be retinotopic and discrete (Schall et al., 1995; Ungerleider et al., 2008). We now also note that to the extent that there are non-retinotopic projections they could indeed be distributed differently onto the functional subtypes of FEF neurons than the retinotopic projections (page 8).

4. In classical working memory tasks, the task periods usually consist of fixation, cue, delay and then response period. The neural activity during delay period is considered as working memory related signal. However, in current study, the authors didn't point out whether only delay period activity was included in analysis when they compared synaptic efficacy between stimulation and non-stimulation trials, in Figure 4a. Because the differences of neuronal response during fixation period cannot be viewed as relevant to information held in working memory, it may be better if only neuronal activity in delay period was included in their analysis.

This analysis of efficacy was indeed limited to the delay period. We regret that this important aspect of the results was not clear, and we have added language to the results (page 9), methods (page 17), and the figure 3 caption to clarify this point.

Relatedly, did all of the 96 visual-recipient FEF neurons show working memory-related activity in the memory-guided saccade task? The example neuron in Figure 3a didn't show a significant difference between In and Out trials during delay period. If the visual-recipient neuron didn't show working memory-related activity, what is the basis for the claim that enhanced synaptic efficacy from V4 to FEF was caused by working memory?

This is an important point of clarification and we thank the reviewer for bringing it up. Indeed, a majority of the visual-recipient neurons did not exhibit elevated delay period activity. As shown in the bar plots in Figure 2b, ~30% of visual-recipient FEF neurons have significant delay activity (no different from the proportion of the non-activated FEF neuronal population). In addition, as mentioned above, we have added an analysis of the relationship between delay-period selectivity and the change in efficacy with WM (Figure 3 Supplement 2). This figure and analysis demonstrates that differences in firing rates between the IN and OUT conditions do not correlate with the change in efficacy.

As for the basis for the claim that enhanced efficacy is a result of WM, that is due to the fact that the location of WM is the only thing that varies between the two behavioral (experimental) conditions between which efficacy is being compared. That is a conclusion that quite reasonably follows from the observed differences in orthodromic efficacy between the two behavioral conditions; the magnitude, latency and synchrony of evoked activity all depend on the location the monkey must remember. Note that we do not claim that efficacy increased specifically in WM-related neurons. On the contrary, in the discussion, we suggest that the increased input to what are preferentially visuomotor neurons (Figure 2) might “facilitate the transformation of visual inputs into motor commands.” This is an important point, but it hardly detracts from the fact that WM deployment (defined behaviorally) clearly alters the efficacy of visual inputs.

5. Did the two example neurons in Figure 4c show adjusted values (subtracting the same measure during non-stimulated trials)? The authors mentioned in Method that Figure 4 showed adjusted values, but it may not be applicable for raster plot in Figure 4c. It may be helpful that using adjusted values show stimulation effects on evoked spike counts during memory In and Out trials.

Note that the raster plots in 3c (formerly 4c) are necessarily unaltered data, as the adjustment procedure does not operate on individual spikes, and the calculation of joint spiking takes into account the relative timing of those individual spikes. We now clarify this in the methods (page 18).

6. Did the authors find some FEF cells showing elevated firing during delay period in outside-RF trials compared with baseline firing? These elevated firings would not be caused by the RF cue and thus could more readily be interpreted as a working memory signal.

This is a good question. Indeed, it has previously been shown that some FEF neurons exhibit visual, motor and delay properties that are not aligned in space (Bruce and Goldberg, 1985; Lawrence et al., 2005). Consistent with this, we found that some neurons exhibited more delay activity when the cue was presented outside of their visual receptive field (values below 0.5 in the plot shown in new Figure 3 Supplement 2)—although the magnitude of their delay selectivity is lower on average than those with elevated firing rates for the IN condition. Importantly, the enhanced efficacy of orthodromic activation occurs in these neurons as well (Δ response magnitude = 0.08±0.01, p<10^-4^, n=26); overall there was no correlation between FEF neurons’ delay selectivity and their change in input efficacy (r = 0.12, p = 0.235, Pearson correlation).

However, the suggestion that elevated activity in the OUT condition is more easily interpreted as reflecting working memory is not correct, in our opinion. One could still argue that such elevated activity is cue induced, e.g. resulting from a release of inhibition. The question of whether delay activity in the FEF, which has been studied and described for decades, is a result of the visual cue vs. memory has been addressed in many ways in previous studies, and we don’t attempt to litigate that issue in the current paper. For example, delay activity often occurs in neurons with no visual response to the cue (e.g. see Figure 4A. in Armstrong et al., 2009), and of course the visual responses of FEF neurons do not persist for the 1 second duration of the delay period (e.g. Sommer and Wurtz, 2001). (We have revised the paper to clarify the duration of the delay period which may have been ambiguous in the original version.)

Importantly, as stated above, the interpretation that the increased efficacy during the delay period results from differences in WM is based solely on the fact that we are comparing efficacy between memory locations during the delay period (e.g., the time during which the monkey must remember the cued location), not because individual FEF neurons display delay activity (or not). Again, the difference in efficacy is independent of the amount of delay activity (Figure 3 Supplement 2). Indeed, the interpretation doesn’t even depend on any presumed role of FEF delay activity in the control of WM.

7. The sample size should be indicated in the Figure 3b Venn diagram.

We have added the sample sizes for non-activated and visual-recipient FEF populations to the Venn diagrams in this figure.

8. It would be useful to indicate the electrical stimulus protocol in Figure 1.

We have moved the stimulation technique, previously shown in Figure 2, into Figure 1, and further details of stimulation timing are now illustrated in Figure 1 supplement 1.

Reviewer #2:This is a very interesting study, examining the properties of different types of neurons in the primate Frontal Eye Fields. It is commonly assumed that a serial processing of information takes place in the frontal lobe, from visual representation, to working memory maintenance, to motor output. However, some evidence to the contrary has also been reported, creating a debate in the field. The authors have characterized meticulously FEF neurons receiving V4 projections, by means of orthodromic stimulation. They report two main findings: that visual-input recipient neurons in FEF exhibit substantial motor activity and that working memory alters the efficacy of V4 input to FEF. The paper provides an important addition to our understanding of FEF processing. Although the first result is unambiguous, and goes against the traditional view of the FEF, the interpretation of the second is less straightforward and would need to be qualified further.

We hope that the revised manuscript provides a clearer picture regarding the significance and interpretation of the second part of the paper. Clarifications and qualifying statements have been added based on reviewer feedback (see responses to points 1-3 below).

1. Orthodromic activation of FEF neurons via V4 stimulation increases the percentage of FEF events that lead to spikes and decreases their latency during working memory. Such an effect appears expectable if FEF neurons are at a higher level of when a stimulus in their receptive field is held in memory compared to a stimulus out of their receptive field. Are the authors suggesting something special about working memory? Would the same outcome not be expected during fixation or smooth pursuit for FEF neurons that are activated by these states? It was not clear that efficacy of transmission itself improves by working memory, just the likelihood that the spiking threshold would be reached.

See response to essential revision #1 above.

2. It would strengthen the authors thesis to discuss the existing functional evidence (in addition to anatomical evidence) that motor FEF neurons receive visual input and can plan movements accordingly. See for example Costello et al. J. Neurosci 2013, 33(41):16394-408.

See response to essential revision #2 above.

3. The authors match the receptive location of FEF and V4 neurons to maximize the chances of identifying monosynaptically connected neurons between the two areas. However, a negative finding of no effects of orthodromic activation does not entirely rule out that the FEF neuron under study receive V4 input, from another site. Some discussion is warranted on this point.

See response to essential revision #3 above.

References:

Armstrong, K. M., Chang, M. H., and Moore, T. (2009). Selection and maintenance of spatial information by frontal eye field neurons. The Journal of Neuroscience, 29(50), 15621–15629. https://doi.org/10.1523/JNEUROSCI.4465-09.2009

Briggs, F., Mangun, G. R., and Usrey, W. M. (2013). Attention enhances synaptic efficacy and the signal-to-noise ratio in neural circuits. Nature, 499(7459), 476–480. https://doi.org/10.1038/nature12276

Bruce, C. J., and Goldberg, M. E. (1985). Primate frontal eye fields. I. Single neurons discharging before saccades. Journal of Neurophysiology, 53(3), 603–635.

Costello, M. G., Zhu, D., Salinas, E., and Stanford, T. R. (2013). Perceptual modulation of motor – But not visual – Responses in the frontal eye field during an urgent-decision task. The Journal of Neuroscience: The Official Journal of the Society for Neuroscience, 33(41), 16394–16408. https://doi.org/10.1523/JNEUROSCI.1899-13.2013

Lawrence, B. M., White, R. L., and Snyder, L. H. (2005). Delay-period activity in visual, visuomovement, and movement neurons in the frontal eye field. Journal of Neurophysiology, 94(2), 1498–1508. https://doi.org/10.1152/jn.00214.2005

Schall, J. D. (1991). Neuronal activity related to visually guided saccades in the frontal eye fields of rhesus monkeys: Comparison with supplementary eye fields. Journal of Neurophysiology, 66(2), 559–579. https://doi.org/10.1152/jn.1991.66.2.559

Schall, J., Morel, A., King, D. J., and Bullier, J. (1995). Topography of visual cortex connections with frontal eye field in macaque: Convergence and segregation of processing streams. The Journal of Neuroscience, 15(6), 4464–4487.

Sommer, M. A., and Wurtz, R. H. (2001). Frontal eye field sends delay activity related to movement, memory, and vision to the superior colliculus. Journal of Neurophysiology, 85(4), 1673–1685.

Ungerleider, L. G., Galkin, T. W., Desimone, R., and Gattass, R. (2008). Cortical connections of area V4 in the macaque. Cerebral Cortex (New York, N.Y.: 1991), 18(3), 477–499. https://doi.org/10.1093/cercor/bhm061